# Review of Advances in Speech Processing with Focus on Artificial Neural Networks

## Douglas O'Shaughnessy

Institut National de la Recherche Scientifique, Quebec, QC G1K 9A9, Canada; dougo@emt.inrs.ca

**Abstract:** Speech is the primary way via which most humans communicate. Computers facilitate this transfer of information, especially when people interact with databases. While some methods to manipulate and interpret speech date back many decades (e.g., Fourier analysis), other processing techniques were developed late last century (e.g., linear predictive coding and hidden Markov models). Nonetheless, the last 25 years have seen major advances leading to the wide acceptance of computer-based speech processing, e.g., cellular telephones and real-time online conversations. This paper reviews older techniques and recent methods that focus largely on artificial neural networks. The major highlights in speech research are examined, without delving into mathematical detail, while giving insight into the research choices that have been made. The focus of this work is to understand how and why the discussed methods function well.

**Keywords:** speech processing; artificial neural networks; spectral analysis; pattern recognition

## 1. Introduction

Speech is the most natural form of human communication. People seek to convey information to others and understand them as well, and vocalizing is convenient and almost universally used for this purpose. In modern times, using computers and artificial intelligence to facilitate many tasks of communication has become common, and speech is among many types of information that are processed efficiently with machines. This paper examines the digital processing of speech for coding (for transmission or storage), recognition (of the content of speech, a speaker's identity, or emotion and health), synthesis (from text), and enhancement purposes. The objective of this paper is to put recent advances in the research on and development of speech methods into perspective, given the historical ways of accomplishing practical speech applications. This is carried out to help explain speech processing without delving into mathematical detail.

## 2. The Nature of Speech Signals

To understand why speech needs to be transformed or "processed" for practical applications, it would be useful to briefly describe how information is represented in the pressure variations that constitute human speech [1]. Natural speech is produced when one exhales air through one's *vocal tract* (VT) while constraining the air's passage through the *vocal cords* (the fleshy tissue in the larynx) and at other points (Figure 1). The typical constriction locations in the VT are the lips (e.g., to make labial consonants), the teeth (for dental fricatives such as /f,v/), the hard upper palate (for so-called alveolar and velar consonants), and the pharynx.

If the vocal cord passage (*glottis)* is adducted to a thin slit, the cords vibrate at a rate called the *fundamental frequency* (F0). This quasi-periodic airflow excites the VT, which acts as a filter to modulate the harmonics (energy in multiples of the F0), creating *voiced* speech. The different shapes of the VT for various *phonemes* (the elemental linguistic sounds of speech) yield spectra with varied resonances that are spaced approximately every 1 kHz (for a typical 17 cm long VT). The center frequencies of these resonances called *formants*

(F1, F2, F3, . . . ) are intentionally varied by speakers to convey the identities of the different phonemes to listeners. Thus, the primary information in speech is found in the values of the formants and F0, as a function of time, as speech dynamically varies. The durations and amplitudes of phoneme sounds are also important for communication, e.g., to help cue syntactic structure and semantic emphasis [2].

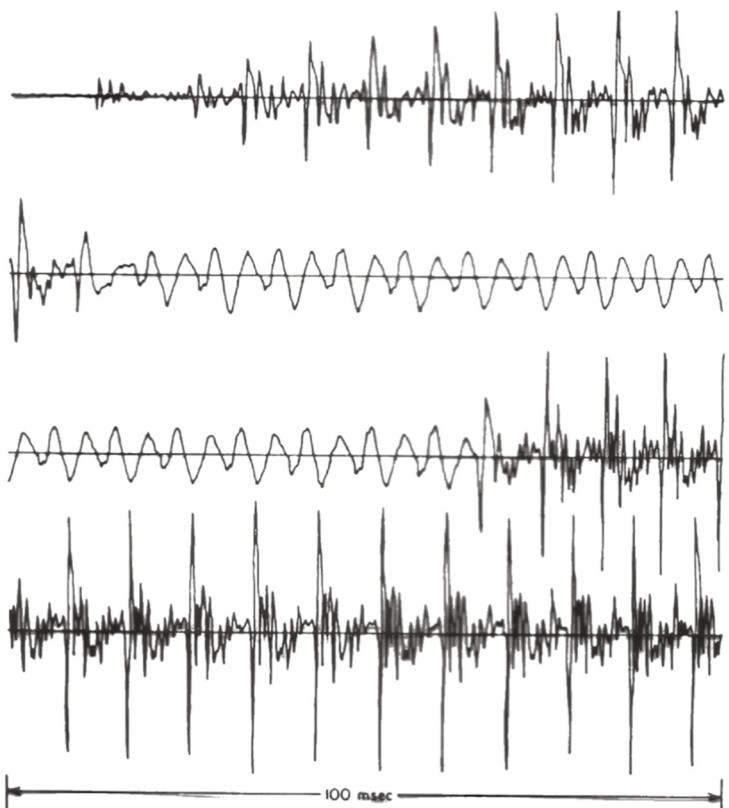

**Figure 1.** Example of a speech time waveform (4 rows in time, each of 100 ms).

Speech has distinct and unique aspects that hinder maximal performance in applications such as automatic recognition and coding. Speech communication evolved over millennia to maximize human survival while accommodating the constraints of both the VT and mammalian hearing mechanisms [3]. Speech likely developed using VT organs that primarily evolved for breathing and eating. While speech perception has much in common with general audition, it is far different from how humans perceive via vision, smell, and touch. To recognize objects in an image, visual processing (whether human- or machine-based) can exploit the intuitive and inherent physical qualities of shapes, textures, and colors. For a video (moving images), one may exploit the constraints imposed by physics (momentum and energy). In contrast, the many levels of encoding found in speech (semantics, syntax, acoustics, phonetics, psychology, and articulation) make speech difficult to process with any single approach.

### 3. Applications of Speech Processing

The earliest artificial use of speech was its electrical transmission via telephone lines. For that, speech processing involved simply converting the pressure variations of speech into electrical currents via a microphone at its source (*encoding*) and later reconstituting audible speech via a loudspeaker or headphones (*decoding*). Vibrating membranes facilitate mechanical/electrical transformations in microphones, loudspeakers, and human eardrums.

In the mid-twentieth century, the advent of digital computers allowed the development of many more applications for speech, as audio representations in binary form could be obtained via *analog-to-digital conversion*. Early digital *telephony coding*, which is still commonly used, sends 8000 signal samples/s with 8 bits/sample (64 kilobits/s). This preserves the 0–4 kHz spectrum, which is the most useful frequency range and is sufficient for most conversations, although it eliminates much of the energy of fricative consonants (while the phonetic context allows listeners to perceive correctly). More advanced coders exploit redundancies in speech production and aspects of audition to lower rates below 10 kilobits/s [4] using processing to be discussed later.

Prior to computers, there were some physical devices, based on musical instruments, that were used to simulate VT airflow and produce *synthetic speech*. Actual *text-to-speech* (TTS), which converts general written messages into intelligible speech, needed computers to transform writing or text characters into phonemes and then emulate VT acoustical behavior [5]. Intelligible and reasonably natural TTS now exists for dozens of languages, although achieving the highest quality often requires much computation beyond the capacity of portable devices.

Since approximately 1960, perhaps the most challenging speech task has been *automatic speech recognition* (ASR), i.e., the conversion of normal spoken utterances into corresponding text [6]. While commercial applications such as Siri, Cortana, and Echo now provide common public service for dozens of languages, human listeners still perform significantly better in difficult acoustic conditions.

Systems can also identify a speaker (*automatic speaker verification* (ASV)), their language, and aspects of their health via speech analysis [7]. Another application of speech processing is for enhancing the quality of degraded speech, e.g., in reverberation and noise [8].

## 4. History of Speech Processing

Early research on speech in the 1940s showed the importance of the distribution of energy in frequency. Analog filter devices, which analyze continuous signals such as speech, created *spectrograms*, which display the intensity of signals as a function of time (horizontal axis) and frequency (vertical axis), wherein a darker shade indicates a higher intensity. Spectrograms are based on a Fourier transform, whereby the bandwidth of pre-chosen filters was typically either narrow (e.g., 45 Hz) or wide (300 Hz). Thus, a narrow-band spectrogram displayed clear harmonics, as speakers rarely have an F0 below 45 Hz, while blurring rapid events in time (e.g., individual vocal cord closures, as the filter response duration exceeded 20 ms). A wide-band spectrogram showed the movements of (formant) resonances, as a 300 Hz width smoothed out harmonics, while preserving brief events, such as plosive consonant releases (Figure 2). The latter is preferred for most applications as resonances and rapid changes are critical for many practical uses.

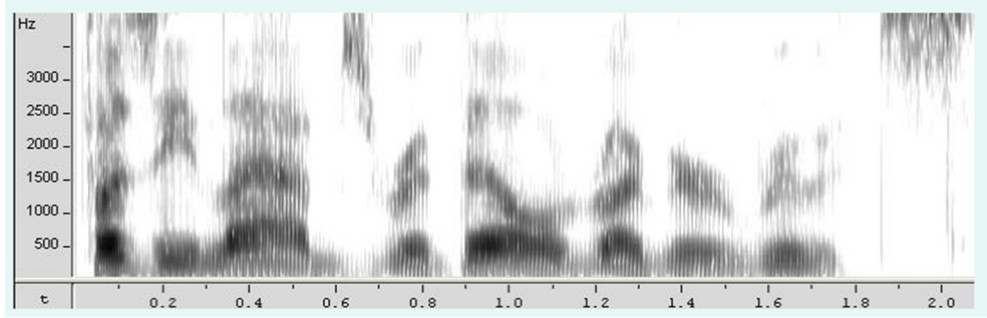

**Figure 2.** Wide-band speech spectrogram. The darkness displays log energy. Vertical: Hz; horizontal: time (s). The (nonsense) text is "Fuzzy labs rebel rugger routes.".

In the 1960s, digital computers facilitated the rapid, precise analysis of many signals, including speech. Persistent advances in computers have allowed great reductions in the

sizes of devices, as well as allowing huge amounts of processing. The Internet has brought a huge increase in available data (both audio and text) to train speech applications. The discussion below highlights the advances in speech applications in recent years, which have been assisted greatly by the progress in computer power and huge amounts of data.

The late 1960s saw a major breakthrough in speech analysis called *linear predictive coding* (LPC) [9]. Earlier work had noted that wide-band speech spectra were dominated by resonances spaced approximately every 1 kHz (for an average adult man; formant spacing is proportionally wider for shorter VTs in women and children). In addition, owing to the relative lowpass nature of air pulses through the glottis, the spectra of voiced speech declined with frequency. This meant that speech waveform samples in each pitch period were generally predictable after each initial excitation at vocal cord closure. As each resonance could be modelled with 2 poles in the complex $z$-plane (of digital analysis), an all-pole spectrum became an efficient model of speech, with 10 poles as the standard for basic telephony (in the 300–3400 Hz range).

The multiplier coefficients of a decoding (synthesis) all-pole filter are readily estimated with the inversion of an *autocorrelation* matrix (obtained directly from the input speech signal via a process similar to convolution) [10]. Early versions of LPC used a simple binary choice of excitation: (1) impulses spaced every F0 for voiced, periodic sounds, and (2) white noise otherwise (Figure 3). In the 1980s, ACELP (algebraic-code-excited LPC) achieved the current quality of cell phones by dedicating the bulk of the 10 kbps transmission to the phase of the LPC residual. (Earlier ADPCM transmitted 3–4 bits/sample of the residual at 32 kbps [4].) Phase has been difficult to exploit in many areas of speech processing as it varies greatly and is not under direct speaker control.

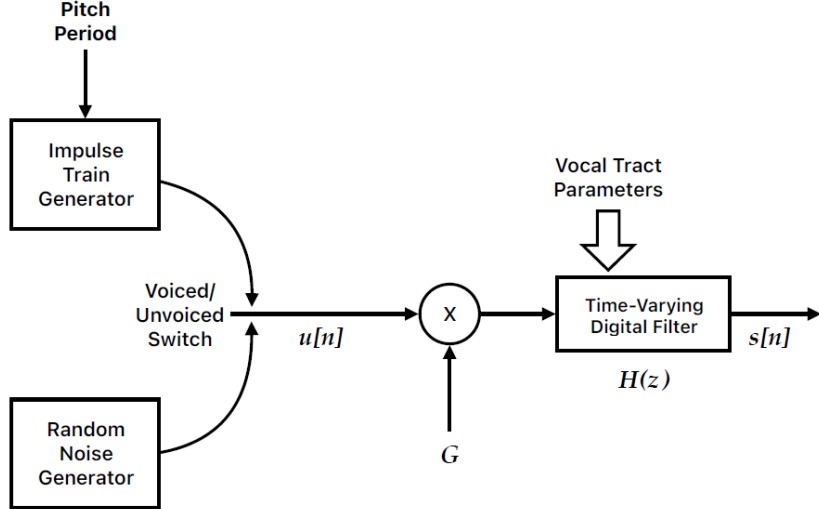

**Figure 3.** Linear predictive coding: all-pole VT filter excited with either white noise or impulses spaced every 1/F0 s.

The use of *mean square error* (MSE) as the criterion to estimate the LPC model simplifies computation but diverges from human perception, which is highly non-linear. As a consequence, in the 1980s, *Mel-frequency cepstral coefficient* (MFCC) analysis became common for speech analysis in speech applications other than coding [11]. MFCC analysis allows the use of frequency warping modelled on the logarithmic behavior of the basilar membrane in the inner ear. Such warping was not feasible for basic LPC for computational reasons. As MFCC analysis uses the Fourier transform (which is not employed in LPC), this allows the deformation of the frequency axis prior to further analysis for ASR, which improves the recognition accuracy. Despite being in common use, MFCC analysis does not exploit the F0 well, averaging over multiple harmonics in high-frequency bands but not at low frequencies. The final inverse-transform step of MFCC analysis has no relation to human perception but is performed to orthogonalize the coefficients.

A variant of LPC called *perceptual linear prediction* (PLP) follows the initial steps of MFCC analysis (FT amplitude and non-uniform frequency-scale weightings) but also adjusts for loudness with a nonlinear model of audition and a modified autocorrelation [12]. PLP allows the use of auditory factors and fewer parameters than in LPC. PLP uses a Bark filter bank of 19 asymmetrically shaped filters, while MFCC analysis typically uses 24–40 triangular filters. These filters approximate the 24 critical bands of human audition and model the logarithmic spacing of hair cells along the inner ear's basilar membrane, which ranges from thin at its basal end (high-frequency filter) to thick at its apex.

Another major breakthrough in speech processing was the use of *hidden Markov models* (HMMs) in the 1980s for ASR [13]. Earlier ASR used *dynamic time warping* (DTW) to compare exemplar templates of test and reference patterns (e.g., of LPC vectors), which was very inefficient for computer memory and computation [14]. Individual templates were used as models for all examples, which provided poor representations without any statistics.

HMMs are able to handle much variability in speech in both time and frequency. Speakers vary greatly in the timing of their articulation, and *coarticulation* (the overlapping VT motion of successive phonemes) greatly affects spectral patterns [15]. However, Markov models assume the first-order independence of frames, which discards much of the useful coarticulation information that listeners exploit in speech understanding.

While not necessarily a breakthrough, spectral sub-bands [16] have long been employed in speech analysis to exploit both the diversity of information across frequencies and the varying human ability to discriminate at different frequencies. Logarithmically spaced bandpass filterbank energies are now commonly used in ASR in place of MFCCs as they are simpler and more flexible to use than MFCCs.

Speech processing often involves data compression, i.e., reducing large numbers of data samples to a much smaller set of information. One form of this is *embeddings*, which are mappings of discrete units of variable duration, such as phonemes, words, or positions, to fixed-length codes [17]. Embeddings carry information about sounds that neighbor other sounds.

This brief history of speech processing prior to the year 2000 sets the stage for the ensuing review of recent advances. Most early applications for speech were dominated by the use of LPC, MFCC analysis, and HMMs. They facilitated the widespread acceptance of cell phones and voice interaction with the internet and via telephone. However, the quality of TTS and the accuracy of ASR remained sub-optimal, leading to further research and development, as discussed below. As noted earlier, the unique challenges of speech (e.g., very indirect encoding in terms of VT resonances and F0 over wide temporal spans) are only partly addressed with the common older techniques.

## 5. Neural Network Advances

The most significant change in speech processing in the last 20 years has been the huge increase in the use of *artificial neural networks* (ANNs). An ANN is a program or process that transforms a sequence of input data samples into an output sequence through a series of layers of nodes, wherein each node receives a sum of weighted values of outputs from the nodes in a prior layer. The output of each node can be binary (zero or one) or an approximation based on a threshold applied to its weighted sum [18]. This mathematical action is loosely similar to that of a biological neuron in the nervous systems of living beings. One distinction is that natural neurons are asynchronous, i.e., each neuron "fires" (raises its output level for a fraction of a millisecond) whenever the weighted sum of its inputs exceeds its threshold, whereas ANNs operate synchronously at a computer cycle rate, running all nodes in a layer at once and then all nodes in the succeeding layer in the next cycle. The threshold makes each node a non-linear processor, which allows ANNs to perform very complex operations. As classifiers, ANNs can create complex decision regions in a representation space (Figure 4).

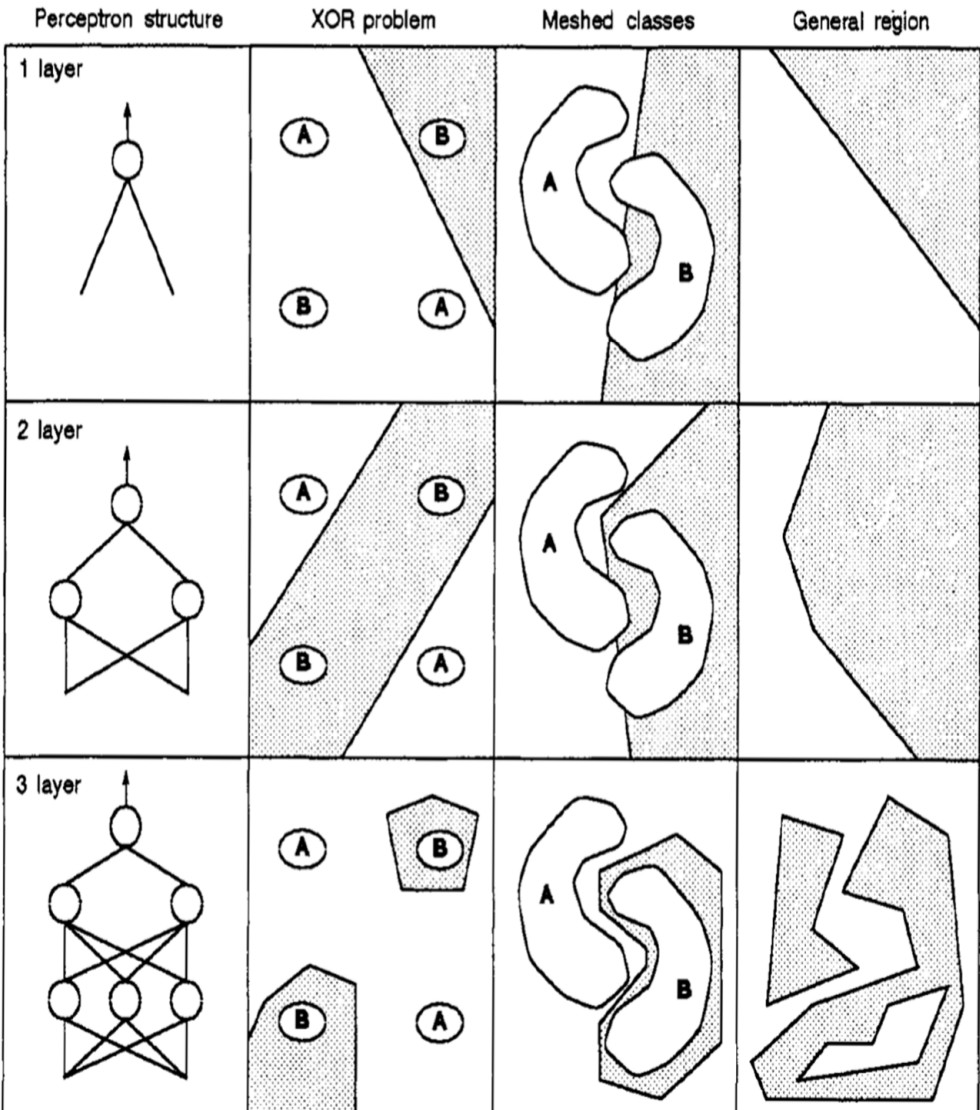

**Figure 4.** Possible decision regions for MLPs (obtained from [18]). First column shows simple 1–6 nodes in 1,2,3-layer model. Second column shows possible decision regions for a 2-class problem (A or B). Third column shows possible regions for 2 classes that may mesh. Fourth column shows examples of possible regions.

### 5.1. Basics of Artificial Neural Networks

The most common application for an ANN is classification, e.g., the recognition of speech or speakers. To estimate the identification of a class for a given input data sequence, a network is trained on an immense number of examples so that the outputs of the final layer of the ANN form a set in which one (desired) node has an output of one and all the others show zero. The objective is for that one node to correspond to the desired class of each input sequence. In the training stage of the network design, the parameters of the ANN (node weights and thresholds) are updated iteratively with an algorithm called *stochastic steepest gradient*, which determines the incremental changes for all parameters by minimizing a cost or error function called a *loss* [19] (Figure 5). In some cases, the loss is directly related to a distortion to minimize elements such as MSE, which can apply for synthesizing or enhancing signals (e.g., in speech coding to compare the input and output of the coder). In many cases, however, the loss is complex and indirect as it must be differentiable to allow the direction (and amount) to be estimated to change the network parameters [20].

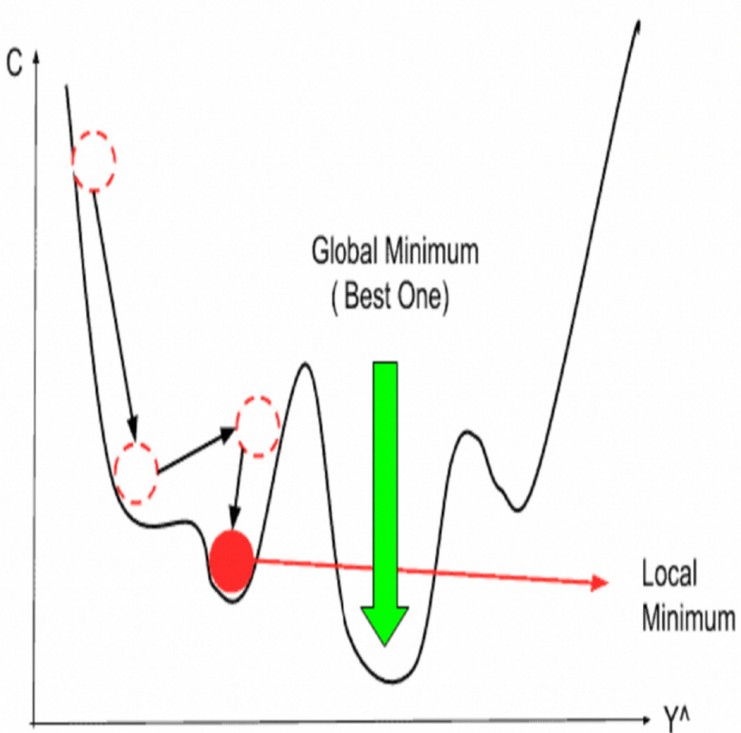

**Figure 5.** Curve demonstrating steepest gradient descent. Given a cost to minimize as a function of network parameters, parameters are adjusted to minimize. As the curve in an ANN is very complex, training is repeated at many different starting points to avoid local minima.

A basic ANN is a fully-connected *multilayer perceptron* (MLP), with a few layers of nodes having all nodes in each layer pass their weighted outputs to each node in the succeeding layer [21]. The inputs of the initial layer are the samples of the data sequence to process, e.g., the successive time samples of a speech signal.

Each node in an ANN processes the sum of its weighted inputs to yield a single output value. Most typically, the process uses an *activation function* to calculate that output [22]. If directly emulating a biological neuron, the node output is one if the sum exceeds the node's threshold and remains at zero otherwise. However, this memoryless function is a discontinuous step and thus not differentiable, blocking the use of a gradient in training. Thus, the chosen activation function for mapping from the weighted inputs to the output is usually a smooth, monotonic function, e.g., tanh, sigmoid, or ReLU (*rectified linear unit*).

The basic ideas for ANNs were developed in the 1960s, but the availability of data, efficient training methods, and computer power were all insufficient for practical speech applications. The last two decades have seen much progress in these needed areas, and we will discuss the resulting advances for speech applications below.

*5.2. Challenges for Artificial Neural Networks*

One weakness of basic ANNs is their complexity in practice. Many applications require inputs of large dimensions, e.g., speech signals consisting of thousands of samples. Modern ANNs require many millions of parameters to provide suitable interpretations of data, as the applications are highly complex. Training these networks with basic techniques that are still in common use (i.e., steepest gradient and simple loss functions) tends to *overfit* the models to the available training data, leading to overly specialized systems that are then less capable to accommodate unexpected inputs. ANNs automatically learn from examples; they are not directed by human expert "advisors." Thus, it may help to seek ways to assist this automatic processing via certain prudent choices of network architecture and find ways to refine or process the input data prior to applying the data to ANNs.

### 5.3. Convolutional Neural Networks (CNNs)

ANNs are trained to automatically seek relevant patterns in data in order to refine or classify aspects of the data. As much progress in ANN development has been driven by the huge field of image processing (e.g., autonomous driving and medical diagnosis) [23], we note the general importance of edges and texture in data to help identify or "enhance" objects in the data. In many applications, important features appear in limited (local) ranges of the data, e.g., the network need not examine data beyond very small ranges. Thus, using fully connected networks is wasteful, and localized analysis can be both efficient and more precise.

CNNs consist of alternating layers of two types: convolutional layers (forming weight sums of inputs) and *pooling* layers (mapping the sums nonlinearly to an output) [24] (Figure 6). Nodes in a convolutional layer receive data from a very limited range called a *receptive field* or *kernel*. Typically, the input is from a two-dimensional matrix, e.g., pixels in an image, and the range is square, e.g., $3 \times 3$. The output can serve to reduce dimensions, e.g., effectively downsample with the value of the range. For example, a $3 \times 3$ kernel outputs one sample, thus performing a 9:1 compression, if the *stride* is three. (A stride of one would simply smooth the data instead.) In CNNs, this pooling occurs in a layer separate from the kernel weighting and can use other functions such as averaging or choosing a maximum value from the kernel.

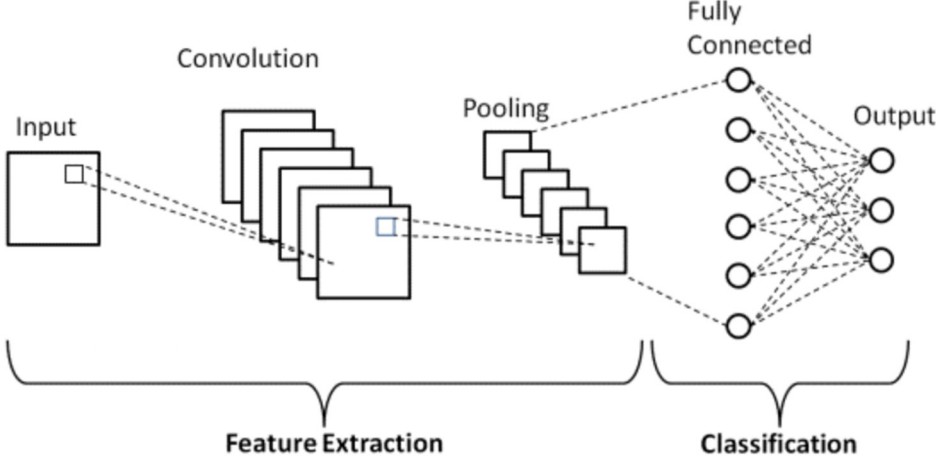

**Figure 6.** Example of a CNN structure.

CNNs try to extract features. Given the typically large set of input data points in most practical applications, CNNs also allow downsampling to reduce dimensionality and, hence, cost while retaining useful patterns. CNNs smooth patterns, reducing incremental variations in the input data that often do not correspond to useful information for the given application (e.g., noisy edges that may reflect imprecision in measuring devices). Many applications apply high sampling rates to accommodate wide bandwidths (e.g., to satisfy the Nyquist rate, which requires twice the highest input frequency [25]), whereas most relevant information often lies at low frequencies; thus, suitable smoothing is often useful.

### 5.4. Recurrent Neural Networks (RNNs)

A second, major recent advancement in ANNs that applies to speech processing is the development of RNNs [26,27]. As noted earlier, general MLPs are often too complex, using thousands of nodes per layer and many layers, with many millions of parameters. For many applications, information (about relevant patterns) in input data is often distributed in a non-uniform fashion in both time and frequency. As a result, most nodes in fully connected networks contribute very little to any specific task. CNNs provide a useful way to process localized patterns, but they cannot handle the many cases wherein relevant patterns extend over very wide ranges.

The use of HMMs for ASR was specifically designed to accommodate correlations in speech over many frames of data, but HMMs have serious weaknesses, as noted earlier. RNNs have a somewhat similar mechanism of incorporating data from different layers of ANNs [20], including recurrent feedback to earlier layers. In basic MLPs, data pass directly to the next layer only, but RNNs allow the passage of data to other layers via the use of specialized *gates*. The states of hidden layers are conditioned on the current input and previous states, which makes the operation recursive. The recurrence in time to update hidden states precludes parallelization, which often blocks the use of RNNs in edge devices.

RNNs accommodate different timings, interpreting words spoken at different speeds as corresponding to the same text. The information in each *cell* in a *long short-term memory* (LSTM) network resides in a state whose inputs move through input gates, which control the entry of data [28]. The state unit has a linear self-loop with weight controlled via a forget gate. The output of each cell can be shut off via an output gate (Figure 7). A reduced version of LSTM called a *gated recurrent unit* is more computationally efficient by using only two gates (vs. three in LSTM) and a single memory cell [29].

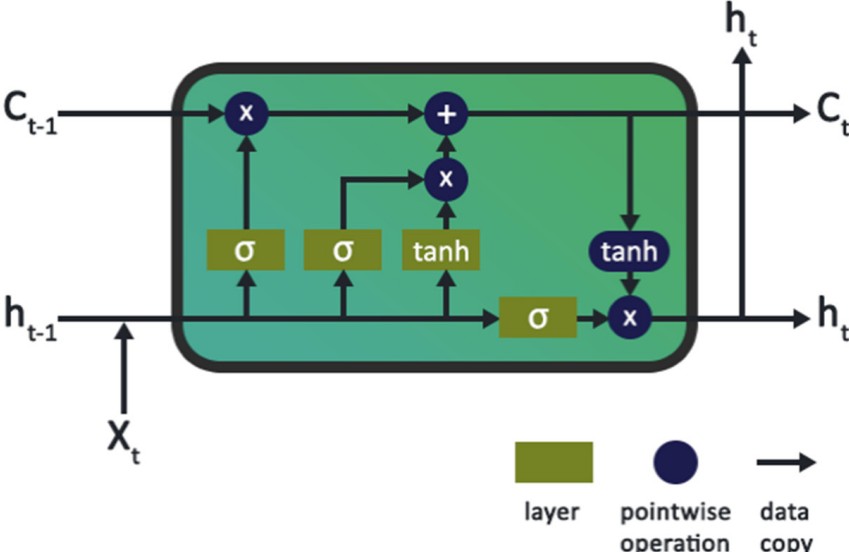

**Figure 7.** Typical LSTM cell showing flow through input, output, and forget gates.

Training ANN parameters usually relies on the derivatives of the loss function, e.g., gradients. Loss gradients are *backpropagated* through the entire ANN [19]. With many layers, this requires long chains of network parameters, wherein effects are the product of many partial derivatives (via the chain rule). Such products can become very small, leading to the *vanishing gradient problem*, whereby relevant effects are lost owing to small values over many network layers [30]. Especially in RNNs, the relevant context may extend over many frames (in acoustic models) or words (in language models). LSTM handles this better than basic RNNs.

*5.5. Attention*

RNNs allow applications to focus on relevant speech segments over a much longer time range than do CNNs or HMMs, but parameter weights in RNNs usually decay over longer ranges of time. RNNs also have as many hidden state vectors as the length of the input sequence, which significantly increases computation. A recent classification method called *attention* [31] allows freedom to choose various data to emphasize to better exploit the distribution of relevant information. It selects aspects in an input sequence (speech/image/text) that are more useful to update internal ANN states to make a prediction for the next output value. Attention is usually interpreted as a *correlation* of relevant

information, and it is calculated via matrix operations (e.g., dot products) that combine several terms: *queries* (inputs), *keys* (features), and *values* (desired outputs weighted via attention), with a *softmax* function to obtain normalized values for attention weights [32]. Parallel (multiple) attention layers are called *heads* [33]. Queries come from an earlier decoder layer, and memory keys and values come from the output of an encoder.

Using correlation to determine relevance in data eases the computation to calculate attention in ANNs, but the choice of correlation is simplistic. Correlation is a first-order linear statistic, and stochastic dependencies are far more complex. It is important to focus on the architecture for efficient neural network operation, not only to reduce cost but also for more accurate performance. Better measures for attention than simple correlation will likely be found in future research.

### 5.6. Transformers

Basic RNNs do not allow parallel processing, which can be crucial for real-time applications. As a result, some recent *end-to-end* (E2E) methods called *transformers* do not use recurrence and rely only on attention to focus on critical aspects of the decision process over a wide range [31]. (E2E means all processing lies in a single uniform ANN, unlike many earlier applications that use a sequential, modular approach with separate acoustic and language models, as well as rescoring.) A transformer has no notion of a token sequence, but instead uses positional encodings in time for data in a separate embedding table. Transformer ASR typically requires more computation than other approaches in terms of optimization, network structure, and data augmentation.

The transformer architecture uses an attention-based encoder and decoder, whereby each module has a stack of identical blocks, each consisting of two sub-layers: a multi-head attention mechanism and a position-wise fully connected feedforward network. Multiple attention heads allow the parts of data sequences to be focused on differently. Speech correlations extend over widely different ranges: very short ranges (e.g., a spectral resonance structure exploited with 10-sample LPC), medium ranges (e.g., the approximate repetition of pitch periods in sonorant phonemes), and long ranges (e.g., F0 behavior over successive words). This complexity makes the application of attention difficult.

### 5.7. Autoencoders

Signal-coding applications use a process of encoding (transforming) an input signal to compress data and then decoding to reconstitute the signal at a receiver. This operation can also apply to signal recognition cases as well, as the data compression can assist in eliminating less pertinent aspects of the data.

An encoder–decoder trained on unlabeled data is called an *autoencoder* [34]. This autoencoder uses a self-supervised encoding step to find data representations (hidden vector representational *encoder embeddings*) in a latent space, while a decoding step is supervised to match the input and output data (the difference, or loss, may be MSE). The encoder often consists of bi- or uni-directional LSTM layers. The *decoding* step generates an output that is as close as possible to the original input (so as to verify that the encoder captures enough feature information in the content to reconstruct the signal with good quality).

### 5.8. Connectionist Temporal Classification (CTC)

A recently developed data processing scheme is called *connectionist temporal classification.* CTC maps an input sequence (e.g., a series of speech frames) to a set of probabilities for all possible corresponding output sequences (e.g., a text of symbols) [35]. An advantage of CTC is its ability to handle the large difference between the number of input frames in many speech applications and the number of output text symbols (e.g., for ASR). A disadvantage of CTC is that it assumes successive output symbols are independent (HMMs also assume strong, conditional independence of input frames, whereas CTC symbols are conditionally independent given the latent state of the neural network, which can depend

on the entire input sequence). CTC allows sequence models to be trained without needing a frame-level alignment of the target labels.

A related approach, but which has an *autoregressive* component, is the *recurrent neural network transducer* [36]. It has a prediction network, but, unlike CTC, it does not assume conditional independence between predictions. (The term "autoregressive" is derived from filtering, wherein recursive (feedback) units in digital filters allow long responses, as in all-pole LPC.)

*5.9. Types of Network Supervision*

Speech system performance is best when networks are trained using labeled data, i.e., *supervised* learning. ANNs have performed well in the supervised learning of high-level feature representations from labeled speech and text data by using layered differentiable models. However, labeled datasets can be costly, and many languages have few available data. *Unsupervised* systems are trained only on speech with no corresponding text and on unspoken text. *Self-supervised learning* (SSL), on the other hand, exploits other information sources [37], while *semi-supervised* methods use some labeled data to create *pseudo-labels* via initial seed models.

5.9.1. Unsupervised Learning

The *Zero Resource Speech Challenge* carries out ASR for five languages with no labeled data. One challenge task uses the unsupervised learning of phoneme-like unit representations, and another task focuses on SSL *spoken term discovery.* One method [38] trains on unlabeled speech to find a mapping to a compact representation (presumably useful features) that helps discriminate between linguistic units (e.g., sub-words such as syllables). The method searches for meaningful word- or phrase-like patterns.

5.9.2. Self-Supervised Learning

SSL pre-trains ANNs on unlabeled data to learn general representations, which are then used to improve the system accuracy with further training on small amounts of labeled data from a target language. Models of lexical learning based on SSL divide data into phonetic units. These models "discover" phonetic features but rarely learn longer-term phonological processes. Averaged latent representations can correspond to relevant phonetic units. SSL clusters speech data automatically into acoustic units that are presumed to share some features relevant to ASR.

For ASR, E2E models convert a variable-length speech input into a set of hidden context states, which are then decoded into a target sequence. Unlike Markov models that limit the range to one prior state, autoregressive methods condition on all past states, i.e., the hidden context states retain information about all prior states.

## 6. Training Data

When examining the reasons for recent major improvements in the performance of speech systems, one must include the burgeoning availability of databases to help train systems. As ANNs are based on examples for training, their accuracy has recently greatly improved with ever-increasing amounts of suitable data. In addition, to help generalize models that are derived directly from examples, *data augmentation* methods have assisted as well [39]. These procedures effectively increase their input training data artificially by perturbing actual data examples in ways that try to retain the relevant aspects of real signals (so that the perturbed data can be viewed as additional training information). Common methods involve randomly omitting portions of signals in time and frequency or adding selective noise and reverberation [40]. As humans tend to be able to correctly perceive such distorted signals (up to a certain point), the view of this research is that such trained models become more robust to mismatches between training and testing data.

### 6.1. Common Databases

Among the early databases of recorded speech are TIMIT, Switchboard, and CallHome. TIMIT contains broadband recordings of 630 speakers of eight major dialects of American English, each reading ten phonetically rich sentences, and they are manually labeled at the phoneme level [41]. Switchboard contains 300 h of training data from telephone conversations between strangers, while CallHome contains 120 conversations between people familiar with each other [42]. The Fisher English Training Speech [43] was developed by the Linguistic Data Consortium and has time-aligned transcript data for 984 h of telephone conversations in English.

A more recent popular read database called LibriSpeech [44] contains a training subset of 360 h of speech from 921 speakers. For Mandarin speech, Aishell is a common dataset [45]. For ASV, VoxCeleb has 2000 h of audio and video of famous people's speech [46].

Recent trends have steered toward larger databases to accommodate the needs of ANNs. While read databases allow more scientific control over experiments (e.g., with common texts to allow more direct comparisons), practical applications are clearly for spontaneous speech, and speaking styles are very different for read versus spontaneous speech.

### 6.2. Model Adaptation

Many speech systems build models to apply in treating speech. These models are based on training, but future testing input often deviates from the prior available training data. To better match trained models to new input, various methods of adaptation have been tried [47]. The many variations in speech signals include speakers, contexts, and acoustic environments.

Model-based adaptation relies on a direct update of ANN parameters. One such method is *speaker-adaptive training* (SAT), which appends speaker-specific *auxiliary features* to network inputs. One typical set is *i-vectors* [48], which can be regarded as basis vectors that span a subspace of speaker variability; they are commonly used for ASV.

### 7. Conclusions

The treatment or processing of speech signals is needed to achieve improved performance of speech applications such as automatic recognition, coding, and enhancement. Basic spectral analysis with the Fourier transform allowed early progress, while timely advances in linear prediction and stochastic modelling led to the increased use of speech systems by the public 40 years ago. The greatly increased recent use of speech applications has been aided in large part by the major advances in artificial neural networks and by the huge increase in computer power and available speech and text data.

**Funding:** This research was funded by the NSERC (grant number 142610).

**Data Availability Statement:** No specific data were used.

**Conflicts of Interest:** The author declares no conflict of interest.

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
