# Peer review of "Review of Advances in Speech Processing with Focus on Artificial Neural Networks"

_electronics, doi:10.3390/electronics12132887_

Round 1

Reviewer 1 Report

This paper summarizes classic information in signal processing and also some advances in neuronal networks.

First of all, the abstract should be improved, the author has to include the advantage of reading this paper instead of others.

Also, the title of this work has to be more specific. That can help readers to find this paper. Recent advances in speech processing focus on artificial neuronal networks.

Sections 1, 2, and 3 describe basic concepts, however, in most of them there are no references. Please cite important concepts.

Following the Electronics template, APA style is not allowed for this journal, Please review Instructions for authors.

This paper should be improved by considering figures, schemes, or graphs for helping readers to understand the methods explained. 

The author has recently published this paper "Review of methods for coding of speech signals". In the opinion of this reviewer, the paper submitted in Electronics should follow a similar structure as the one already published.

Author Response

>This paper summarizes classic information in signal processing and also some advances in neuronal networks.  First of all, the abstract should be improved, the author has to include the advantage of reading this paper instead of others.

I revised the abstract to include this information.

>Also, the title of this work has to be more specific. That can help readers to find this paper. Recent advances in speech processing focus on artificial neuronal networks.

I altered the title accordingly.

>Sections 1, 2, and 3 describe basic concepts, however, in most of them there are no references. Please cite important concepts.

I added significant citations, and cite important concepts.  There are 8 citations now in these sections.

>Following the Electronics template, APA style is not allowed for this journal, Please review Instructions for authors.

I revised the format accordingly.

>This paper should be improved by considering figures, schemes, or graphs for helping readers to understand the methods explained. 

I added 7 figures.

>The author has recently published this paper "Review of methods for coding of speech signals". In the opinion of this reviewer, the paper submitted in Electronics should follow a similar structure as the one already published.

I revised this paper accordingly.

Reviewer 2 Report

I assume that this review article is intended to be an introduction to the Electronics special issue on Recent Advances in Audio, Speech and Music Processing and Analysis. In that case, it should  provide  explanations for the technology whixh are coherent but do not require specialist knowledge. The article reads as if there was a size limit: some sections are abrupt.

There are mistakes in the manuscript and some small edits to the English are required. I am attaching a marked-up version identifying these. More substantial points are covered below.

The article has no figures and some are badly needed: to illustrate spectrograms and ANN architectures for instance.

line 130: Shouldn't PLP get a mention here?

An important motivation for using MFCCs is that the features are roughly orthogonal, making mixture Gaussian modelling more appropriate. You can contrast that with ANNs later.

line 140: the fundamental limitation of DTW is that one example stands for all the articulations of a speech unit, and cannot represent the natural variation. That's why you need models.

line 190 In early expts with ANNs there were no training theorems.

line 213 figure needd to explain convolutional layer.. best with image example

line 220 explanation of weights should come earlier

l248 explanation of RNNs difficult to understand, doesnt explain the R!

From about this point there are few references.

l265 Attention explanation introduces a lot of terms with specialist meanings which won't be understood.. queries, heads, keys, values

There are mistakes in the manuscript and some small edits to the English are required. I am attaching a marked-up version identifying these

Author Response

>I assume that this review article is intended to be an introduction to the Electronics special issue on Recent Advances in Audio, Speech and Music Processing and Analysis. In that case, it should  provide  explanations for the technology which are coherent but do not require specialist knowledge. The article reads as if there was a size limit: some sections are abrupt.

I have expanded sections to augment the material, toward a goal as specified here by the reviewer.  (By the way, this article has not specifically been targeted as an introduction to the special issue, as far as I know.)

>There are mistakes in the manuscript and some small edits to the English are required. I am attaching a marked-up version identifying these. More substantial points are covered below. 

I apologize for these errors, and have fixed them in the revised text.

>The article has no figures and some are badly needed: to illustrate spectrograms and ANN architectures for instance.

I have included serval relevant figures in the revised version, including a spectrogram and ANN architectures.

>line 130: Shouldn't PLP get a mention here?

Yes, I added that.  As R2 noted earlier, my first first version was too succinct.  I have now expanded.

>An important motivation for using MFCCs is that the features are roughly orthogonal, making mixture Gaussian modelling more appropriate. You can contrast that with ANNs later.

Yes, I added this information.

>line 140: the fundamental limitation of DTW is that one example stands for all the articulations of a speech unit, and cannot represent the natural variation. That's why you need models.

Yes, I expanded my text to include this point.  Earlier, I had not made explicit this important point about the limitations of exemplar-based DTW.

>line 190 In early expts with ANNs there were no training theorems.

Yes, that was indeed a serious limitation, improved vastly in recent years.  I added this note.

>line 213 figure needed to explain convolutional layer.. best with image example

I added a figure for CNN, but do not wish to give an image example in a speech paper.

>line 220 explanation of weights should come earlier

Yes, agreed, and and done.

>l248 explanation of RNNs difficult to understand, doesnt explain the R!  From about this point there are few references.

I was indeed too brief about RNN in the first version, and have added much more.  I added references.

>l265 Attention explanation introduces a lot of terms with specialist meanings which won't be understood.. queries, heads, keys, values

I agree, and added more detail.

Reviewer 3 Report

The topic of neural network advances in speech processing, in particular deep learning, is of significant interest and importance.  However, this paper merely touches upon some of the neural network techniques without offering the reader insight into these techniques. Even without equations, side by side comparisons of performance and/or improvements from one algorithm to the other could have been provided in tables. 

More importantly, there are recently published considerably more comprehensive reviews, such as the following, that may provide the author examples for a state-of-the-art reviews:

Mehrish, Ambuj, Navonil Majumder, Rishabh Bharadwaj, Rada Mihalcea, and Soujanya Poria. "A review of deep learning techniques for speech processing." Information Fusion (2023): 101869.

Bhangale, Kishor Barasu, and Mohanaprasad Kothandaraman. "Survey of deep learning paradigms for speech processing." Wireless Personal Communications 125, no. 2 (2022): 1913-1949. 

Author Response

>The topic of neural network advances in speech processing, in particular deep learning, is of significant interest and importance.  However, this paper merely touches upon some of the neural network techniques without offering the reader insight into these techniques. Even without equations, side by side comparisons of performance and/or improvements from one algorithm to the other could have been provided in tables. 

I have revised the paper significantly to aid in offering more insight.

>More importantly, there are recently published considerably more comprehensive reviews, such as the following, that may provide the author examples for a state-of-the-art reviews:

Mehrish, Ambuj, Navonil Majumder, Rishabh Bharadwaj, Rada Mihalcea, and Soujanya Poria. "A review of deep learning techniques for speech processing." Information Fusion (2023): 101869.

Bhangale, Kishor Barasu, and Mohanaprasad Kothandaraman. "Survey of deep learning paradigms for speech processing." Wireless Personal Communications 125, no. 2 (2022): 1913-1949. 

I greatly thank the reviewer for pointing these two articles out to me; I was unaware of them, as they are not in mainstream speech/language journals.  These two articles present far too much detail to be useful as a review for novices.  Instead, they may serve as useful summaries of the state-of-the-art for speech research experts, which is not my target readership.

While the Mehresh article is interesting, it has many poor expressions.  For example, it states: “in signal processing, a signal that repetitively manifests after a fixed duration, known as a period, is classified as periodic. The reciprocal of this period represents the frequency of the signal.” This is very awkwardly phrased, and clearly wrong; signals generally have a broad spectrum, not one frequency.  The text then continues: “The waveform of a periodic signal defines its shape and concurrently determines its timbre, which pertains to the subjective perception of sound quality by humans.”  What is “shape” here?  Defining timbre as subjective perception of sound quality is very vague.  These are just two examples of many poor phrasings.  Many terms are simply used without explanation, e.g., query, key, value, head, regularization.

Its choice of citations is very suspect: “time-domain and frequency-domain features tend to capture different sets of information and thus can be used in conjunction to solve a task [16–18].” - This is very simplistic and uses a strange melange of references in terms of age and application.

Figure 1 in the Mehresh article starts at 2000, and ignores the major application of speech coding (except to mention LPC).  My work examines advances before 2000, and includes speech coding. Their section 2.3 conflates models with classification methods.  The level of insight in the Mehresh article is less, I believe, than mine, as it has numerous statements such as “Deep reinforcement learning ..can learn from raw data without needing hand-engineered features, making it more flexible and adaptable. It can also learn from feedback, making it more robust and able to handle noisy environments.”  Such statements deserve much more justification than is seen.

As for the Bhangale article, it is much less well written, and it too is more detailed (20K words) than my much briefer article (7K words).  It tends to list an exhaustive series of works in the field, with inadequate organization or summaries to help the reader understand.  The paper shows one after another of methods, with little evaluation or comparison for insight.  It is also full of grammatical errors.

Both articles say little about the essential features of speech signals, and are much more oriented around details of methods, rather than motivating how and why these methods were developed.  They tend to overwhelm the reader with so much detail, without focussing on wha

Reviewer 4 Report

The article, although named "recent advances in speech processing", uses more then 1/3 of the space by general description and history. I would expect more detailed description recent techniques perhaps their comparison, advantages and weaknesses.

  Since authors did not want to delve into equations, so there is just a vague description of the techniques, not even illustrated by the images. Sometimes, the description is given without clear reason (e.g.  gradient vanishing problem) and outcomes.

  I certainly would appreciate more citations. For sure for the original work and then some follow-ups and  improvements to direct the readers to  specific research.

Also information should be technically correct  - the telephone line is 300 - 3400 HZ (lines 67, 115), first spectrograms were not displaying intensity in grey-scale, but were similar to seismograph and plotted the amplitudes. 

Also MFCC were not designed to handle F0 and the final inverse transform has different reason and goal then emulate human hearing.

In section 5.3. are described activation functions, but there are used in all ANNs, also in MLPs.

lines 234 - 237 discuss higher sampling rate, but the reasoning and conclusions are weird.

"For speaker adaptation, model probability distributions may be normalized to zero mean and unit variance, prior to applying inputs to an ANN."  --  this does not make sense. Probability of which modes are normalized?

line 404 -- Patents -- what is the purpose?

There are some minor, but frequent, mistakes in English such us missing verbs or wrong tense (e.g. line 358 :... are derive directly ..." --> derived; line 379 "The recent trend has been towards..." -- hes been directed)

Author Response

>The article, although named "recent advances in speech processing", uses more than 1/3 of the space by general description and history. I would expect more detailed description recent techniques perhaps their comparison, advantages and weaknesses.

I have changed the title, omitting “recent”, as it is important to place advances in historical context.  I  have added material to better show comparison, advantages and weaknesses.

>  Since authors did not want to delve into equations, so there is just a vague description of the techniques, not even illustrated by the images. Sometimes, the description is given without clear reason (e.g.  gradient vanishing problem) and outcomes.

I have added figures, and amplified on method descriptions.

>  I certainly would appreciate more citations. For sure for the original work and then some follow-ups and  improvements to direct the readers to  specific research.

I have added many more citations.

>Also information should be technically correct  - the telephone line is 300 - 3400 HZ (lines 67, 115), 

I changed this, although different sources have varying published ranges.

>first spectrograms were not displaying intensity in grey-scale, but were similar to seismograph and plotted the amplitudes. 

I changed this.

>Also MFCC were not designed to handle F0 and the final inverse transform has different reason and goal then emulate human hearing.

I modified the description here, to better explain.

>In section 5.3. are described activation functions, but there are used in all ANNs, also in MLPs.

Yes, I had placed this paragraph poorly, and it is now moved earlier.

>lines 234 - 237 discuss higher sampling rate, but the reasoning and conclusions are weird.

I modified the description here, to better explain.

>"For speaker adaptation, model probability distributions may be normalized to zero mean and unit variance, prior to applying inputs to an ANN."  --  this does not make sense. Probability of which modes are normalized? 

I deleted that short section.

>line 404 -- Patents -- what is the purpose?

I do not cite patents in the paper.

>Comments on the Quality of English Language

There are some minor, but frequent, mistakes in English such us missing verbs or wrong tense (e.g. line 358 :... are derive directly ..." --> derived; line 379 "The recent trend has been towards..." -- has been directed)

I apologize for these errors, now corrected.  I have gone through the paper more thoroughly.

Round 2

Reviewer 1 Report

Good improvement, please check the size of figures 5 and 6.

Author Response

>Good improvement, please check the size of figures 5 and 6.

my submitted figures as requested by the journal were in a zip file; we will make sure these figures are the correct size.

Reviewer 2 Report

Thank you for paying attention to the points I raised. Them/s is much improved but a few minor problems remain:

Figure 1 caption :  '...a speech time waveform, with three phonemes...'. No, the phoneme is a perceived unit: it's what you hear. You'd have to say 'perceived as 3 phonemes', but I wonder if having a waveform is worthwhile at all.

Figure 2: better say what's being spoken.

line 175 better say that embeddings carry information about the sounds which are neighbouring other sounds.

Figure 4 needs explanation

Figure 5 scale is distorted.. possibly a pdf problem

Figure 6 similar

English is OK now.

Author Response

Thank you for paying attention to the points I raised. Them/s is much improved but a few minor problems remain:

Figure 1 caption :  '...a speech time waveform, with three phonemes...'. No, the phoneme is a perceived unit: it's what you hear. You'd have to say 'perceived as 3 phonemes', but I wonder if having a waveform is worthwhile at all.

>I have changed the legend accordingly.  I would be OK with deleting this figure, but the first round of reviews specifically asked for more figures, and the other 3 reviewers did not ask to delete this.

Figure 2: better say what's being spoken.

>I added the text to the legend.

line 175 better say that embeddings carry information about the sounds which are neighbouring other sounds.

> I added that point.

Figure 4 needs explanation

> I added that to the legend; also, the image should have been rotated clockwise (the journal inserted it off-angle)

Figure 5 scale is distorted.. possibly a pdf problem

Figure 6 similar

>Yes, my submitted figures as requested by the journal were in a zip file; we will make sure these figures are the correct size.

English is OK now.

> thanks.

Reviewer 3 Report

Thank you for addressing my original concerns. I agree with the revised edition.

Author Response

> thank you for addressing my original concerns. I agree with the revised edition.

Thanks.